# From Hard to Soft: Understanding Deep Network Nonlinearities via Vector Quantization and Statistical Inference

**Randall Balestriero & Richard G. Baraniuk**
Department of Electrical and Computer Engineering
Rice University
Houston, TX 77005, USA
`randallbalestriero@gmail.com`

## Abstract

Nonlinearity is crucial to the performance of a deep (neural) network (DN). To date there has been little progress understanding the menagerie of available nonlinearities, but recently progress has been made on understanding the rôle played by piecewise affine and convex nonlinearities like the ReLU and absolute value activation functions and max-pooling. In particular, DN layers constructed from these operations can be interpreted as *max-affine spline operators* (MASOs) that have an elegant link to vector quantization (VQ) and $K$-means. While this is good theoretical progress, the entire MASO approach is predicated on the requirement that the nonlinearities be piecewise affine and convex, which precludes important activation functions like the sigmoid, hyperbolic tangent, and softmax. *This paper extends the MASO framework to these and an infinitely large class of new nonlinearities by linking deterministic MASOs with probabilistic Gaussian Mixture Models (GMMs).* We show that, under a GMM, piecewise affine, convex nonlinearities like ReLU, absolute value, and max-pooling can be interpreted as solutions to certain natural "hard" VQ inference problems, while sigmoid, hyperbolic tangent, and softmax can be interpreted as solutions to corresponding "soft" VQ inference problems. We further extend the framework by hybridizing the hard and soft VQ optimizations to create a $\beta$-VQ inference that interpolates between hard, soft, and linear VQ inference. A prime example of a $\beta$-VQ DN nonlinearity is the *swish* nonlinearity, which offers state-of-the-art performance in a range of computer vision tasks but was developed ad hoc by experimentation. Finally, we validate with experiments an important assertion of our theory, namely that DN performance can be significantly improved by enforcing orthogonality in its linear filters.

## 1 Introduction

Deep (neural) networks (DNs) have recently come to the fore in a wide range of machine learning tasks, from regression to classification and beyond. A DN is typically constructed by composing a large number of linear/affine transformations interspersed with up/down-sampling operations and simple scalar nonlinearities such as the ReLU, absolute value, sigmoid, hyperbolic tangent, etc. Goodfellow et al. (2016). Scalar nonlinearities are crucial to a DN's performance. Indeed, without nonlinearity, the entire network would collapse to a simple affine transformation. *But to date there has been little progress understanding and unifying the menagerie of nonlinearities, with few reasons to choose one over another other than intuition or experimentation.*

Recently, progress has been made on understanding the rôle played by *piecewise affine and convex nonlinearities* like the ReLU, leaky ReLU, and absolute value activations and downsampling operations like max-, average-, and channel-pooling Balestriero & Baraniuk (2018a;b). In particular, these operations can be interpreted as *max-affine spline operators* (MASOs) Magnani & Boyd (2009); Hannah & Dunson (2013) that enable a DN to find a locally optimized piecewise affine approximation to the prediction operator given training data. A spline-based prediction is made in

two steps. First, given an input signal $\boldsymbol{x}$, we determine which region of the spline's partition of the domain (the input signal space) it falls into. Second, we apply to $\boldsymbol{x}$ the fixed (in this case affine) function that is assigned to that partition region to obtain the prediction $\widehat{y} = f(\boldsymbol{x})$.

The key result of Balestriero & Baraniuk (2018a;b) is *any DN layer constructed from a combination of linear and piecewise affine and convex is a MASO*, and hence the entire DN is merely a composition of MASOs.

MASOs have the attractive property that their partition of the signal space (the collection of multi-dimensional "knots") is completely determined by their affine parameters (slopes and offsets). This provides an elegant link to *vector quantization* (VQ) and $K$-*means clustering*. That is, during learning, a DN implicitly constructs a hierarchical VQ of the training data that is then used for spline-based prediction.

This is good progress for DNs based on ReLU, absolute value, and max-pooling, but what about DNs based on classical, high-performing nonlinearities that are neither piecewise affine nor convex like the sigmoid, hyperbolic tangent, and softmax or fresh nonlinearities like the *swish* Ramachandran et al. (2017) that has been shown to outperform others on a range of tasks?

**Contributions.** In this paper, we address this gap in the DN theory by developing a new framework that unifies a wide range of DN nonlinearities and inspires and supports the development of new ones. *The key idea is to leverage the yinyang relationship between deterministic VQ/$K$-means and probabilistic Gaussian Mixture Models (GMMs)* Biernacki et al. (2000). Under a GMM, piecewise affine, convex nonlinearities like ReLU and absolute value can be interpreted as solutions to certain natural *hard inference* problems, while sigmoid and hyperbolic tangent can be interpreted as solutions to corresponding *soft inference* problems. We summarize our primary contributions as follows:

Contribution 1: We leverage the well-understood relationship between VQ, $K$-means, and GMMs to propose the *Soft MASO* (SMASO) model, a probabilistic GMM that extends the concept of a deterministic MASO DN layer. Under the SMASO model, *hard maximum a posteriori (MAP) inference* of the VQ parameters corresponds to conventional deterministic MASO DN operations that involve piecewise affine and convex functions, such as fully connected and convolution matrix multiplication; ReLU, leaky-ReLU, and absolute value activation; and max-, average-, and channel-pooling. These operations assign the layer's input signal (feature map) to the VQ partition region corresponding to the closest centroid in terms of the Euclidean distance,

Contribution 2: A hard VQ inference contains no information regarding the confidence of the VQ region selection, which is related to the distance from the input signal to the region boundary. In response, we develop a method for *soft MAP inference* of the VQ parameters based on the probability that the layer input belongs to a given VQ region. *Switching from hard to soft VQ inference recovers several classical and powerful nonlinearities and provides an avenue to derive completely new ones.* We illustrate by showing that the soft versions of ReLU and max-pooling are the sigmoid gated linear unit and softmax pooling, respectively. We also find a home for the sigmoid, hyperbolic tangent, and softmax in the framework as a new kind of DN layer where the MASO output is the VQ probability.

Contribution 3: We generalize hard and soft VQ to what we call $\beta$-*VQ inference*, where $\beta \in (0, 1)$ is a free and learnable parameter. This parameter interpolates the VQ from linear ($\beta \to 0$), to probabilistic SMASO ($\beta = 0.5$), to deterministic MASO ($\beta \to 1$). We show that the $\beta$-VQ version of the hard ReLU activation is the *swish* nonlinearity, which offers state-of-the-art performance in a range of computer vision tasks but was developed ad hoc through experimentation Ramachandran et al. (2017).

Contribution 4: Seen through the MASO lens, current DNs solve a simplistic per-unit (per-neuron), independent VQ optimization problem at each layer. In response, we extend the SMASO GMM to a *factorial GMM* that that supports jointly optimal VQ across all units in a layer. Since the factorial aspect of the new model would make naïve VQ inference exponentially computationally complex, *we develop a simple sufficient condition under which a we can achieve efficient, tractable, jointly optimal VQ inference.* The condition is that the linear "filters" feeding into any nonlinearity should be *orthogonal*. We propose two simple strategies to learn approximately and truly orthogonal weights and show on three different datasets that both offer significant improvements in classification per-

formance. Since orthogonalization can be applied to an arbitrary DN, this result and our theoretical understanding are of independent interest.

This paper is organized as follows. After reviewing the theory of MASOs and VQ for DNs in Section 2, we formulate the GMM-based extension to SMASOs in Section 3. Section 4 develops the hybrid $\beta$-VQ inference with a special case study on the swish nonlinearity. Section 5 extends the SMASO to a factorial GMM and shows the power of DN orthogonalization. We wrap up in Section 6 with directions for future research. Proofs of the various results appear in several appendices in the Supplementary Material.

## 2 BACKGROUND ON MAX-AFFINE SPLINES AND DEEP NETWORKS

We first briefly review *max-affine spline operators* (MASOs) in the context of understanding the inner workings of DNs Balestriero & Baraniuk (2018a;b). A MASO is an operator $S[A, B] : \mathbb{R}^D \to \mathbb{R}^K$ that maps an input vector of length $D$ into an output vector of length $K$ by concatenating $K$ independent *max-affine splines* Magnani & Boyd (2009); Hannah & Dunson (2013), with each spline formed from $R$ piecewise affine and convex mappings. The MASO parameters consist of the "slopes" $A \in \mathbb{R}^{K \times R \times D}$ and the "offsets/biases" $B \in \mathbb{R}^{K \times R}$. See Appendix A for the precise definition. Given the input $\boldsymbol{x} \in \mathbb{R}^D$ and parameters $A, B$, a MASO produces the output $\boldsymbol{z} \in \mathbb{R}^K$ via

$$[\boldsymbol{z}]_k = [S[A, B](\boldsymbol{x})]_k = \max_{r=1,\dots,R} \left( \langle [A]_{k,r,.}, \boldsymbol{x} \rangle + [B]_{k,r} \right), \tag{1}$$

where $[\boldsymbol{z}]_k$ denotes the $k^{\text{th}}$ dimension of $\boldsymbol{z}$. The three subscripts of the slopes tensor $[A]_{k,r,d}$ correspond to output $k$, partition region $r$, and input signal index $d$. The two subscripts of the offsets/biases tensor $[B]_{k,r}$ correspond to output $k$ and partition region $r$.

An important consequence of (1) is that a MASO is completely determined by its slope and offset parameters without needing to specify the partition of the input space (the "knots" when $D = 1$). Indeed, solving (1) automatically computes an optimized partition of the input space $\mathbb{R}^D$ that is equivalent to a *vector quantization* (VQ) Nasrabadi & King (1988); Gersho & Gray (2012). We can make the VQ aspect explicit by rewriting (1) in terms of the *Hard-VQ* (HVQ) matrix $T_H \in \mathbb{R}^{K \times R}$. that contains $K$ stacked one-hot row vectors, each with the one-hot position at index $[\boldsymbol{t}]_k \in \{1, \dots, R\}$ corresponding to the $\arg \max$ over $r = 1, \dots, R$ of (1). Given the HVQ matrix, (or equivalently, a region of the input space), the input-output mapping is affine and fully determined by

$$[\boldsymbol{z}]_k = \sum_{r=1}^{R} [T_H]_{k,r} \left( \langle [A]_{k,r,.}, \boldsymbol{x} \rangle + [B]_{k,r} \right). \tag{2}$$

We retrieve (1) from (2) by noting that $[\boldsymbol{t}]_k = \arg \max_{r=1,\dots,R}(\langle [A]_{k,r,.}, \boldsymbol{x} \rangle + [B]_{k,r})$.

The key background result for this paper is that the *layers* of a very large class of DN are MASOs. Hence, such a DN is a composition of MASOs, where each layer MASO has as input the feature map $\boldsymbol{z}^{(\ell-1)} \in \mathbb{R}^{D^{(\ell-1)}}$ and produces $\boldsymbol{z}^{(\ell)} \in \mathbb{R}^{D^{(\ell)}}$, with $\ell$ corresponding to the layer. Each MASO has thus specific parameters $A^{(\ell)}, B^{(\ell)}$.

**Theorem 1.** *Any DN layer comprising a linear operator (e.g., fully connected or convolution) composed with a convex and piecewise affine operator (such as a ReLU, leaky-ReLU, or absolute value activation; max/average/channel-pooling; maxout; all with or without skip connections) is a MASO Balestriero & Baraniuk (2018a;b).*

Appendix A provides the parameters $A^{(\ell)}, B^{(\ell)}$ for the MASO corresponding to the $\ell^{\text{th}}$ layer of any DN constructed from linear plus piecewise affine and convex components. Given this connection, we will identify $\boldsymbol{z}^{(\ell-1)}$ above as the input (feature map) to the MASO DN layer and $\boldsymbol{z}^{(\ell)}$ as the output (feature map). We also identify $[\boldsymbol{z}^{(\ell)}]_k$ in (1) and (2) as the output of the $k^{\text{th}}$ *unit* (aka neuron) of the $\ell^{\text{th}}$ layer. MASOs for higher-dimensional tensor inputs/outputs are easily developed by flattening.

## 3 MAX-AFFINE SPLINES MEET GAUSSIAN MIXTURE MODELS

The MASO/HVQ connection provides deep insights into how a DN clusters and organizes signals layer by layer in a hierarchical fashion Balestriero & Baraniuk (2018a;b). However, the entire ap-

proach requires that the nonlinearities be piecewise affine and convex, which precludes important activation functions like the sigmoid, hyperbolic tangent, and softmax. *The goal of this paper is to extend the MASO analysis framework of Section 2 to these and an infinitely large class of other nonlinearities by linking deterministic MASOs with probabilistic Gaussian Mixture Models (GMMs).*

## 3.1 FROM MASO TO GMM VIA $K$-MEANS

For now, we focus on a single unit $k$ from layer $\ell$ of a MASO DN, which contains both linear and nonlinear operators; we generalize below in Section 5. The key to the MASO mechanism lies in the VQ variables $[\boldsymbol{t}^{(\ell)}]_k \; \forall k$, since they fully determine the output via (2). For a special choice of bias, the VQ variable computation is equivalent to the $K$-*means* algorithm Balestriero & Baraniuk (2018a;b).

**Proposition 1.** *Given* $-\frac{1}{2}\left\|\left[A^{(\ell)}\right]_{k,r,\cdot}\right\|^2 = \left[B^{(\ell)}\right]_{k,r}$, *the MASO VQ partition corresponds to a $K$-means clustering[1] with centroids* $\left[A^{(\ell)}\right]_{k,r,\cdot}$ *computed via* $[\widehat{\boldsymbol{t}^{(\ell)}}]_k = \underset{r=1,\ldots,R}{\arg\min} \left\|\left[A^{(\ell)}\right]_{k,r,\cdot} - \boldsymbol{z}^{(\ell-1)}\right\|^2$.

For example, consider a layer $\ell$ using a ReLU activation function. Unit $k$ of that layer partitions its input space using a $K$-means model with $R^{(\ell)} = 2$ centroids: the origin of the input space and the unit layer parameter $[A^{(\ell)}]_{k,1,\cdot}$. The input is mapped to the partition region corresponding to the closest centroid in terms of the Euclidean distance, and the corresponding affine mapping for that region is used to project the input and produce the layer output as in (2).

We now leverage the well-known relationship between $K$-means and Gaussian Mixture Models (GMMs) Bishop (2006) to GMM-ize the deterministic VQ process of max-affine splines. As we will see, the constraint on the value of $\left[B^{(\ell)}\right]_{k,r}$ in Proposition 1 will be relaxed thanks to the GMM's ability to work with a nonuniform prior over the regions (in contrast to $K$-means).

To move from a deterministic MASO model to a probabilistic GMM, we reformulate the HVQ selection variable $[\boldsymbol{t}^{(\ell)}]_k$ as an unobserved categorical variable $[\boldsymbol{t}^{(\ell)}]_k \sim \mathcal{C}at([\pi^{(\ell)}]_{k,\cdot})$ with parameter $[\pi^{(\ell)}]_{k,\cdot} \in \triangle_{R^{(\ell)}}$ and $\triangle_{R^{(\ell)}}$ the simplex of dimension $R^{(\ell)}$. Armed with this, we define the following generative model for the layer input $\boldsymbol{z}^{(\ell-1)}$ as a mixture of $R^{(\ell)}$ Gaussians with mean $[A^{(\ell)}]_{k,r,\cdot} \in \mathbb{R}^{D^{(\ell-1)}}$ and identical isotropic covariance with parameter $\sigma^2$

$$\boldsymbol{z}^{(\ell-1)} = \sum_{r=1}^{R^{(\ell)}} \mathbb{1}\left([\boldsymbol{t}^{(\ell)}]_k = r\right) \left[A^{(\ell)}\right]_{k,r,\cdot} + \epsilon, \tag{3}$$

with $\epsilon \sim \mathcal{N}(0, I\sigma^2)$. Note that this GMM generates an independent vector input $\boldsymbol{z}^{(\ell-1)}$ for every unit $k = 1,\ldots,D^{(\ell)}$ in layer $\ell$. For reasons that will become clear below in Section 3.3, we will refer to the GMM model (3) as the *Soft MASO* (SMASO) model. We develop a joint, factorial model for the entire MASO layer (and not just one unit) in Section 5.

## 3.2 HARD VQ INFERENCE

Given the GMM (3) and an input $\boldsymbol{z}^{(\ell-1)}$, we can compute a *hard inference* of the optimal VQ selection variable $[\boldsymbol{t}^{(\ell)}]_k$ via the maximum a posteriori (MAP) principle

$$[\widehat{\boldsymbol{t}^{(\ell)}}]_k = \underset{t=1,\ldots,R^{(\ell)}}{\arg\max} \; p(t|\boldsymbol{z}^{(\ell-1)}). \tag{4}$$

The following result is proved in Appendix E.1.

**Theorem 2.** *Given a GMM with parameters* $\sigma^2 = 1$ *and* $[\pi^{(\ell)}]_{k,t} = \frac{\exp([B^{(\ell)}]_{k,r} + \frac{1}{2}\|[A^{(\ell)}]_{k,r,\cdot}\|^2)}{\sum_r \exp([B^{(\ell)}]_{k,r} + \frac{1}{2}\|[A^{(\ell)}]_{k,r,\cdot}\|^2)}$, $t = 1,\ldots,R^{(\ell)}$, *the MAP inference of the latent selection variable* $[\boldsymbol{t}^{(\ell)}]_k$ *given in (4) can be computed via the MASO HVQ (1)*

$$[\widehat{\boldsymbol{t}^{(\ell)}}]_k = \underset{r=1,\ldots,R^{(\ell)}}{\arg\max} \left\langle [A^{(\ell)}]_{k,t,\cdot}, \boldsymbol{z}^{(\ell-1)}\right\rangle + [B^{(\ell)}]_{k,t}, \qquad \forall A^{(\ell)} \; \forall B^{(\ell)}. \tag{5}$$

---

[1] It would be more accurate to call this $R^{(\ell)}$-means clustering in this case.

*The optimal HVQ selection matrix is given by* $[\widehat{T_H^{(\ell)}}]_{k,r} = \mathbb{1}\big(r = [\widehat{\boldsymbol{t}^{(\ell)}}]_k\big).$

Note in Theorem 2 that the bias constraint of Proposition 1 (which can be interpreted as imposing a uniform prior $[\pi^{(\ell)}]_{k,\cdot}$) is completely relaxed.

HVQ inference of the selection matrix sheds light on some of the drawbacks that affect any DN employing piecewise affine, convex activation functions. First, during gradient-based learning, the gradient will propagate back only through the activated VQ regions that correspond to the few 1-hot entries in $T_H^{(\ell)}$. The parameters of other regions will not be updated; this is known as the "dying neurons phenomenon" Trottier et al. (2017); Agarap (2018). Second, the overall MASO mapping is continuous but not differentiable, which leads to unexpected gradient jumps during learning. Third, the HVQ inference contains no information regarding the confidence of the VQ region selection, which is related to the distance of the query point to the region boundary. As we will now see, this extra information can be very useful and gives rise to a range of classical and new activation functions.

## 3.3 SOFT VQ INFERENCE

We can overcome many of the limitations of HVQ inference in DNs by replacing the 1-hot entries of the HVQ selection matrix with the probability that the layer input belongs to a given VQ region

$$[\widehat{T_S^{(\ell)}}]_{k,r} = p\left([t^{(\ell)}]_k = r \mid \boldsymbol{z}^{(\ell-1)}\right) = \frac{\exp\left(\big\langle[A^{(\ell)}]_{k,r,\cdot}, \boldsymbol{z}^{(\ell-1)}\big\rangle + [B^{(\ell)}]_{k,r}\right)}{\sum_r \exp\left(\big\langle[A^{(\ell)}]_{k,r,\cdot}, \boldsymbol{z}^{(\ell-1)}\big\rangle + [B^{(\ell)}]_{k,r}\right)}, \quad (6)$$

which follows from the simple structure of the GMM. This corresponds to a *soft inference* of the categorical variable $[\boldsymbol{t}^{(\ell)}]_k$. Note that $T_S^{(\ell)} \to T_H^{(\ell)}$ as the noise variance in (3) $\to 0$. Given the SVQ selection matrix, the MASO output is still computed via (2). The SVQ matrix can be computed indirectly from an entropy-penalized MASO optimization; the following is reproved in Appendix E.2 for completeness.

**Proposition 2.** *The entries of the SVQ selection matrix* $[\widehat{T_S^{(\ell)}}]_{k,\cdot}$ *from (6) solve the following entropy-penalized maximization, where* $H(\cdot)$ *is the Shannon entropy*[2]

$$[\widehat{T_S^{(\ell)}}]_{k,\cdot} = \underset{t \in \triangle_{R_k^{(\ell)}}}{\arg\max} \sum_{r=1}^{R_k^{(\ell)}} [t]_r \left(\big\langle[A^{(\ell)}]_{k,r,\cdot}, \boldsymbol{z}^{(\ell-1)}\big\rangle + [B^{(\ell)}]_{k,r}\right) + H(t). \quad (7)$$

Proposition 2, which was first established in Manning & Klein (2003); Mount (2011), unifies HVQ and SVQ in a single optimization problem. The transition from HVQ (5) to SVQ (7) is obtained simply by adding the entropy regularization $H(t)$. Notice that removing the Entropy regularization from (7) leads to the same VQ as (5). We summarize this finding in Table. 1.

## 3.4 SOFT VQ MASO NONLINEARITIES

Remarkably, switching from HVQ to SVQ MASO inference recovers several classical and powerful nonlinearities and provides an avenue to derive completely new ones. Given a set of MASO parameters $A^{(\ell)}, B^{(\ell)}$ for calculating the layer-$\ell$ output of a DN via (1), we can derive two distinctly different DNs: one based on the HVQ inference of (5) and one based on the SVQ inference of (6). The following results are proved in Appendix E.5.

**Proposition 3.** *The MASO parameters* $A^{(\ell)}, B^{(\ell)}$ *that induce the ReLU activation under HVQ induce the* sigmoid gated linear unit *Elfwing et al. (2018) under SVQ.*

**Proposition 4.** *The MASO parameters* $A^{(\ell)}, B^{(\ell)}$ *that induce the max-pooling nonlinearity under HVQ induce* softmax-pooling *Boureau et al. (2010) under SVQ.*

Appendix C discusses how the GMM and SVQ formulations shed new light on the impact of parameter initialization in DC learning plus how these formulations can be extended further.

---

[2]The observant reader will recognize this as the E-step of the GMM's EM learning algorithm.

| VQ Type | Value for $[T^{(\ell)}]_k$ | Examples |
|---|---|---|
| Hard VQ (HVQ) | $\arg\max_{t\in\triangle_{R_k^{(\ell)}}} \mathcal{P}(t)$ | ReLU, max-pooling |
| Soft VQ (SVQ) | $\arg\max_{t\in\triangle_{R_k^{(\ell)}}} \mathcal{P}(t) + H(t)$ | SiGLU, softmax-pooling |
| $\beta$-VQ, $\beta \in [0,1]$ | $\arg\max_{t\in\triangle_{R_k^{(\ell)}}} \beta\mathcal{P}(t) + (1-\beta)H(t)$ | swish, $\beta$-softmax-pooling |

Table 1: Impact of different VQ strategies for a MASO layer with $\mathcal{P}(t) := \sum_{r=1}^{R_k^{(\ell)}} [t]_r \left( \langle [A^{(\ell)}]_{k,r,.}, \boldsymbol{z}^{(\ell-1)} \rangle + [B^{(\ell)}]_{k,r} \right)$.

### 3.5 Additional Nonlinearities as Soft DN Layers

Changing viewpoint slightly, we can also derive classical nonlinearities like the sigmoid, tanh, and softmax Goodfellow et al. (2016) from the soft inference perspective. Consider a new *soft DN layer* whose unit output $[\boldsymbol{z}^{(\ell)}]_k$ is not the piecewise affine spline of (2) but rather the probability $[\boldsymbol{z}^{(\ell)}]_k = p([\boldsymbol{t}^{(\ell)}]_k = 1|\boldsymbol{z}^{(\ell-1)})$ that the input $\boldsymbol{z}^{(\ell)}$ falls into each VQ region. The following propositions are proved in Appendix E.6.

**Proposition 5.** *The MASO parameters $A^{(\ell)}, B^{(\ell)}$ that induce the ReLU activation under HVQ induce the* sigmoid *activation in the corresponding soft DN layer.*[3]

A similar train of thought recovers the softmax nonlinearity typically used at the DN output for classification problems.

**Proposition 6.** *The MASO parameters $A^{(\ell)}, B^{(\ell)}$ that induce a fully-connected-pooling layer under HVQ (with output dimension $D^{(L)}$ equal to the number of classes $C$) induce the softmax nonlinearity in the corresponding soft DN layer.*

## 4 Hybrid Hard/Soft Inference via Entropy Regularization

Combining (5) and (6) yields a hybrid optimization for a new $\beta$-*VQ* that recovers hard, soft, and linear VQ inference as special cases

$$[\widehat{T_\beta^{(\ell)}}]_k = \arg\max_{t\in\triangle_{R_k^{(\ell)}}} [\beta^{(\ell)}]_k \sum_{r=1}^{R_k^{(\ell)}} [t]_r \left( \langle [A^{(\ell)}]_{k,r,.}, \boldsymbol{z}^{(\ell-1)} \rangle + [B]_{k,r} \right) + \left(1 - [\beta^{(\ell)}]_k\right) H(t), \quad (8)$$

with the new hyper-parameter $[\beta^{(\ell)}]_k \in (0,1)$. The $\beta$-VQ obtained from the above optimization problem utilizes $[\beta^{(\ell)}]_k$ to balance the impact of the regularization term (introduced in the SVQ derivation (7)), allowing to recover and interpolate the VQ between linear, soft and hard (see Table 1). The following is proved in Appendix E.3.

**Theorem 3.** *The unique global optimum of (8) is given by*

$$[\widehat{T_\beta^{(\ell)}}]_{k,r} = \frac{\exp\left( \frac{[\beta^{(\ell)}]_k}{1-[\beta^{(\ell)}]_k} \left( \langle [A^{(\ell)}]_{k,r}, \boldsymbol{z}^{(\ell-1)} \rangle + [B^{(\ell)}]_{k,r} \right) \right)}{\sum_{j=1}^{R^{(\ell)}} \exp\left( \frac{[\beta^{(\ell)}]_k}{1-[\beta^{(\ell)}]_k} \left( \langle [A^{(\ell)}]_{k,j,.}, \boldsymbol{z}^{(\ell-1)} \rangle + [B^{(\ell)}]_{k,j} \right) \right)}. \quad (9)$$

The $\beta$-VQ covers all of the theory developed above as special cases: $\beta = 1$ yields HVQ, $\beta = \frac{1}{2}$ yields SVQ, and $\beta = 0$ yields a linear MASO with $[\widehat{T_0^{(\ell)}}]_{k,r} = \frac{1}{R^{(\ell)}}$. See Figure 1 for examples of how the $\beta$ parameter interacts with three example activation functions. Note also the attractive property that (9) is differentiable with respect to $[\beta^{(\ell)}]_k$.

The $\beta$-VQ supports the development of new, high-performance DN nonlinearities. For example, the *swish activation* $\sigma_{\text{swish}}(u) = \sigma_{\text{sig}}([\eta^{(\ell)}]_k u)u$ extends the sigmoid gated linear unit Elfwing et al. (2018) with the learnable parameter $[\eta^{(\ell)}]_k$ Ramachandran et al. (2017). Numerous experimental studies have shown that DNs equipped with a learned swish activation significantly outperform those with more classical activations like ReLU and sigmoid.[4]

---

[3] The tanh activation is obtained similarly by reparametrizing $A^{(\ell)}$ and $B^{(\ell)}$; see Appendix E.6.

[4] Best performance was usually achieved with $[\eta^{(\ell)}]_k \in (0,1)$ Ramachandran et al. (2017).

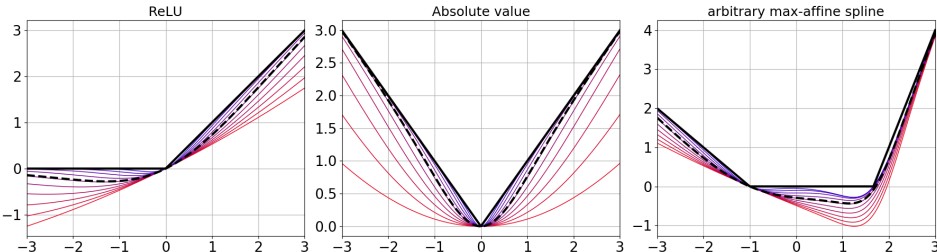

Figure 1: For the MASO parameters $A^{(\ell)}, B^{(\ell)}$ for which HVQ yields the ReLU, absolute value, and an arbitrary convex activation function, we explore how changing $\beta$ in the $\beta$-VQ alters the induced activation function. Solid black: HVQ ($\beta = 1$), Dashed black: SVQ ($\beta = \frac{1}{2}$), Red: $\beta$-VQ ($\beta \in [0.1, 0.9]$). Interestingly, note how some of the functions are nonconvex.

**Proposition 7.** *The MASO $A^{(\ell)}, B^{(\ell)}$ parameters that induce the ReLU nonlinearity under HVQ induce the* swish *nonlinearity under $\beta$-VQ, with $[\eta^{(\ell)}]_k = \frac{[\beta^{(\ell)}]_k}{1 - [\beta^{(\ell)}]_k}$.*

Table 1 summarizes some of the many nonlinearities that are within reach of the $\beta$-VQ.

## 5  OPTIMAL JOINT VQ INFERENCE VIA ORTHOGONALIZATION

The GMM (3) models the impact of only a single layer unit on the layer-$\ell$ input $z^{(\ell-1)}$. We can easily extend this model to a *factorial model* for $z^{(\ell-1)}$ that enables all $D^{(\ell)}$ units at layer $\ell$ to combine their syntheses:

$$z^{(\ell-1)} = \sum_{k=1}^{D^{(\ell)}} \sum_{r=1}^{R^{(\ell)}} \mathbb{1}\left([t^{(\ell)}]_k = r\right) [A^{(\ell)}]_{k,r,\cdot} + \epsilon, \tag{10}$$

with $\epsilon \sim \mathcal{N}(0, I\sigma^2)$. This new model is a mixture of $R^{(\ell)}$ Gaussians with means $[A^{(\ell)}]_{k,r,\cdot} \in \mathbb{R}^{D^{(\ell-1)}}$ and identical isotropic covariances with variance $\sigma^2$. The factorial aspect of the model means that the number of possible combinations of the $t^{(\ell)}$ values grow exponentially with the number of units. Hence, inferring the latent variables $t^{(\ell)}$ quickly becomes intractable.

However, we can break this combinatorial barrier and achieve efficient, tractable VQ inference by constraining the MASO slope parameters $A^{(\ell)}$ to be orthogonal

$$\left\langle [A^{(\ell)}]_{k,r,\cdot}, [A^{(\ell)}]_{k',r',\cdot} \right\rangle = 0 \quad \forall k \neq k' \; \forall r, r'. \tag{11}$$

Orthogonality is achieved in a fully connected layer (multiplication by the dense matrix $W^{(\ell)}$ composed with activation or pooling) when the rows of $W^{(\ell)}$ are orthogonal. Orthogonality is achieved in a convolution layer (multiplication by the convolution matrix $C^{(\ell)}$ composed with activation or pooling) when the rows of $C^{(\ell)}$ are either non-overlapping or properly apodized; see Appendix E.4 for the details plus the proof of the following result.

**Theorem 4.** *If the slope parameters $A^{(\ell)}$ of a MASO are orthogonal in the sense of (11), then the random variables $[t^{(\ell)}]_1|z^{(\ell-1)}, \ldots, [t^{(\ell)}]_1|z^{(\ell-1)}$ of the model (10) are independent and hence $p\left([t^{(\ell)}]_1, \ldots, [t^{(\ell)}]_{D^{(\ell)}}|z^{(\ell-1)}\right) = \prod_{k=1}^{D^{(\ell)}} p\left([t^{(\ell)}]_k|z^{(\ell-1)}\right).$*

In an orthogonal, factorial MASO, optimal inference can be performed independently per factor, as opposed to jointly over all of the factors. Orthogonality renders the joint MAP inference of the factorial model's VQs tractable. The following result is proved in Appendix E.4.

Practically, this not only lowers the computational complexity tremendously but also imparts the benefit of "uncorrelated unit firing," which has been shown to be advantageous in DNs Srivastava et al. (2014). Beyond the scope of this paper, such an orthogonalization strategy can also be applied to more general factorial models such as factorial GMMs Zemel (1994); Ghahramani (1995) and factorial HMMs Ghahramani & Jordan (1996).

| Setting | $LR = 0.001$ | $LR = 0.0005$ | $LR = 0.0001$ |
|---|---|---|---|
| SVHN (baseline) | $94.3 \pm 0.1$ | $94.4 \pm 0.1$ | $93.4 \pm 0.0$ |
| SVHN Ortho | $94.6 \pm 0.2$ | $95.0 \pm 0.2$ | $93.8 \pm 0.1$ |
| CIFAR10 (baseline) | $80.3 \pm 0.4$ | $80.2 \pm 0.2$ | $76.2 \pm 0.3$ |
| CIFAR10 Ortho | $84.0 \pm 0.3$ | $82.3 \pm 0.1$ | $79.1 \pm 0.2$ |
| CIFAR100 (baseline) | $43.6 \pm 0.2$ | $44.1 \pm 0.4$ | $37.5 \pm 0.5$ |
| CIFAR100 Ortho | $46.1 \pm 0.2$ | $46.3 \pm 0.2$ | $42.1 \pm 0.3$ |

Table 2: Classification experiment to demonstrate the utility of orthogonal DN layers. For three datasets and the same *largeCNN* architecture (detailed in Appendix D), we tabulate the classification accuracy (larger is better) and its standard deviation averaged over 5 runs with different Adam learning rates. In each case, orthogonal fully-connected and convolution matrices improve the classification accuracy over the baseline.

**Corollary 1.** *When the conditions of Theorem 4 are fulfilled, the joint MAP estimate for the VQs of the factorial model (10)*

$$\widehat{\boldsymbol{t}_f^{(\ell)}} = \underset{t \in \{1,...,R^{(\ell)}\} \times \cdots \times \{1,...,R^{(\ell)}\}}{\arg\max} p\left(t | \boldsymbol{z}^{(\ell-1)}\right) = \left[[\widehat{\boldsymbol{t}^{(\ell)}}]_1, \ldots, [\widehat{\boldsymbol{t}^{(\ell)}}]_{D^{(\ell)}}\right]^{\mathsf{T}} \qquad (12)$$

*and thus can be computed with linear complexity in the number of units.*

The advantages of orthogonal or near-orthogonal filters have been explored empirically in various settings, from GANs Brock et al. (2016) to RNNs Huang et al. (2017), typically demonstrating improved performance. Table 2 tabulates the results of a simple confirmation experiment with the *largeCNN* architecture described in Appendix D. We added to the standard cross-entropy loss a term $\lambda \sum_k \sum_{k' \neq k} \sum_{r,r'} \langle [A^{(\ell)}]_{k,r,\cdot}, [A^{(\ell)}]_{k',r',\cdot} \rangle^2$ that penalizes non-orthogonality (recall (11)). We did not cross-validate the penalty coefficient $\lambda$ but instead set it equal to 1. The tabulated results show clearly that favoring orthogonal filters improves accuracy across both different datasets and different learning settings.

Since the orthogonality penalty does not guarantee true orthogonality but simply favors it, we performed one additional experiment where we reparametrized the fully-connected and convolution matrices using the Gram-Schmidt (GS) process Daniel et al. (1976) so that they were truly orthogonal. Thanks to the differentiability of all of the operations involved in the GS process, we can backpropagate the loss to the orthogonalized filters in order to update them in learning. We also used the swish activation, which we showed to be a $\beta$-VQ nonlinearity in Section 4. Since the GS process adds significant computational overhead to the learning algorithm, we conducted only one experiment on the largest dataset (CIFAR100). The exactly orthogonalized *largeCNN* achieved a classification accuracy of $61.2\%$, which is a major improvement over all of the results in the bottom (CIFAR100) cell of Table 2. This indicates that there are good reasons to try to improve on the simple orthogonality-penalty-based approach.

## 6 FUTURE WORK

Our development of the SMASO model opens the door to several new research questions. First, we have merely scratched the surface in the exploration of new nonlinear activation functions and pooling operators based on the SVQ and $\beta$-VQ. For example, the soft- or $\beta$-VQ versions of leaky-ReLU, absolute value, and other piecewise affine and convex nonlinearities could outperform the new swish nonlinearity. Second, replacing the entropy penalty in the (7) and (8) with a different penalty will create entirely new classes of nonlinearities that inherit the rich analytical properties of MASO DNs. Third, orthogonal DN filters will enable new analysis techniques and DN probing methods, since from a signal processing point of view problems such as denoising, reconstruction, compression have been extensively studied in terms of orthogonal filters. This work was partially supported by NSF grants IIS-17-30574 and IIS-18-38177, AFOSR grant FA9550-18-1-0478, ARO grant W911NF-15-1-0316, ONR grants N00014-17-1-2551 and N00014-18-12571, DARPA grant G001534-7500, and a DOD Vannevar Bush Faculty Fellowship (NSSEFF) grant N00014-18-1-2047.

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

# SUPPLEMENTARY MATERIALS

## A    BACKGROUND

A Deep Network (DN) is an operator $f_\Theta : \mathbb{R}^D \to \mathbb{R}^C$ that maps an input signal $\boldsymbol{x} \in \mathbb{R}^D$ to an output prediction $y \in \mathbb{R}^C$. All current DNs can be written as a composition of $L$ intermediate mappings called *layers*

$$f_\Theta(\boldsymbol{x}) = \left( f_{\theta^{(L)}}^{(L)} \circ \cdots \circ f_{\theta^{(1)}}^{(1)} \right)(\boldsymbol{x}), \tag{13}$$

where $\Theta = \left\{ \theta^{(1)}, \ldots, \theta^{(L)} \right\}$ is the collection of the network's parameters from each layer. The DN layer at level $\ell$ is an operator $f_{\theta^{(\ell)}}^{(\ell)}$ that takes as input the vector-valued signal $\boldsymbol{z}^{(\ell-1)}(\boldsymbol{x}) \in \mathbb{R}^{D^{(\ell-1)}}$ and produces the vector-valued output $\boldsymbol{z}^{(\ell)}(\boldsymbol{x}) \in \mathbb{R}^{D^{(\ell)}}$ with $D^{(L)} = C$. The signals $\boldsymbol{z}^{(\ell)}(\boldsymbol{x}), \ell > 1$ are typically called *feature maps* an the input is denoted as $\boldsymbol{z}^{(0)}(\boldsymbol{x}) = \boldsymbol{x}$. For concreteness, we will focus here on processing multi-channel images $x$ but adjusting the appropriate dimensionalities can be used to adapt our results. We will use two equivalent representations for the signal and feature maps, one based on tensors and one based on flattened vectors. In the *tensor* representation, $z^{(\ell)}$ contains $C^{(\ell)}$ channels of size $\left( I^{(\ell)} \times J^{(\ell)} \right)$ pixels. In the *vector* representation, $[\boldsymbol{z}^{(\ell)}(\boldsymbol{x})]_k$ represents the entry of the $k^{\text{th}}$ dimension of the flattened, vector version $\boldsymbol{z}^{(\ell)}(\boldsymbol{x})$ of $z^{(\ell)}(\boldsymbol{x})$. Hence, $D^{(\ell)} = C^{(\ell)} I^{(\ell)} J^{(\ell)}$, $C^{(L)} = C$, $I^{(L)} = 1$, and $J^{(L)} = 1$. For conciseness we will often denote $\boldsymbol{z}^{(\ell)}(\boldsymbol{x})$ as $\boldsymbol{z}^{(\ell)}$. When using nonlinearities and pooling which are piecewise affine and convex, the layers and whole DN fall under the analysis of max-affine spline operators (MASOs) developed in Balestriero & Baraniuk (2018a). In this framework, a *max-affine spline operator* with parameters $A^{(\ell)} \in \mathbb{R}^{D^{(\ell)} \times R \times D^{(\ell-1)}}$ and $B^{(\ell)} \in \mathbb{R}^{D^{(\ell)} \times R}$ is defined as

$$\boldsymbol{z}^{(\ell)} = S\Big[A^{(\ell)}, B^{(\ell)}\Big](\boldsymbol{z}^{(\ell-1)}) = \begin{bmatrix} \max_{r=1,\ldots,R}\langle [A^{(\ell)}]_{1,r,.}, \boldsymbol{z}^{(\ell-1)} \rangle + [B^{(\ell)}]_{1,r} \\ \vdots \\ \max_{r=1,\ldots,R}\langle [A^{(\ell)}]_{K,r,.}, \boldsymbol{z}^{(\ell-1)} \rangle + [B^{(\ell)}]_{K,r} \end{bmatrix}. \tag{14}$$

Any DN layer made of convex and piecewise affine nonlinearities or pooling can be rewritten exactly as a MASO. Hence, such operators take place of the layer mappings of (13) We first proceed by modifying (14) to highlight the internal inference problem. We first introduce the *VQ-matrix* $T^{(\ell)} \in \mathbb{R}^{D^{(\ell)} \times R}$ which will be used to make the mapping region specific, as in

$$A^{(\ell)}[T^{(\ell)}] = \begin{bmatrix} (\sum_{r=1}^{R}[T^{(\ell)}]_{1,r}[A^{(\ell)}]_{1,r,.})^T \\ \vdots \\ (\sum_{r=1}^{R}[T^{(\ell)}]_{K,r}[A^{(\ell)}]_{K,r,.})^T \end{bmatrix}, B^{(\ell)}[T^{(\ell)}] = \begin{bmatrix} (\sum_{r=1}^{R}[T^{(\ell)}]_{1,r}[B^{(\ell)}]_{1,r,.})^T \\ \vdots \\ (\sum_{r=1}^{R}[T^{(\ell)}]_{K,r}[B^{(\ell)}]_{K,r,.})^T \end{bmatrix}, \tag{15}$$

effectively making $A^{(\ell)}[T^{(\ell)}]$ a matrix of shape $(D^{(\ell)}, D^{(\ell-1)})$ and $B^{(\ell)}[T^{(\ell)}]$ a vector of length $D^{(\ell)}$. Hence the VQ-matrix is used to combined the per region parameters. In a standard MASO, each row of $T^{(\ell)}$ is a one-hot vector at position corresponding to the region in which the input falls into. Due to the one-hot encoding present in $T^{(\ell)}$ we refer to this inference as a hard-VQ.

**Proposition 8.** *For a MASO, the VQ-matrix is denoted as $T_H^{(\ell)}$ and is obtained via the internal maximization process of (14). It corresponds to the (hard-)VQ of the input. Once computed the output is a simple affine transform of the input as*

$$\boldsymbol{z}^{(\ell)} = A^{(\ell)}[T_H^{(\ell)}]\boldsymbol{z}^{(\ell-1)} + B^{(\ell)}[T_H^{(\ell)}]. \tag{16}$$

*with* $[T_H^{(\ell)}]_{k,r} = \mathbb{1}_{\{r=\arg\max_{r=1,\ldots,R}\langle [A^{(\ell)}]_{k,r,.}, \boldsymbol{z}^{(\ell-1)} \rangle + [B^{(\ell)}]_{k,r}\}}$.

The VQ matrix $T_H^{(\ell)}$ always belongs to the set of all matrices with different one-hot positions (from 1 to $R$) for each of the output dimensions $k = 1, \ldots, D^{(\ell)}$. We denote this VQ-matrix space as $\mathcal{T}_H^{(\ell)} = \{[a_1, \ldots, a_{D^{(\ell)}}]^T, a_k \in \{\boldsymbol{e}_1, \ldots, \boldsymbol{e}_R\}\}$ with $\boldsymbol{e}_r = \delta_r, dim(\boldsymbol{e}_r) = R$.

## B    ORTHOGONAL FILTERS DETAILS

The developed results on orthogonality induce orthogonality of the case of fully-connected layers. For the case on convolutional layer it implies orthogonality as well as non overlapping patches. This is not practical as it considerably reduces the spatial dimensions making very deep network unsuitable. As such we now

propose a brief approximation result. Due to the specificity of the convolution operator we are able to provide a tractable inference coupled with an apodization scheme. To demonstrate this, we first highlight that any input can be represented as a direct sum of its apodized patches. Then, we see that filtering apodized patches with a filter is equivalent to convolving the input with apodized filters. We first need to introduce the patch notation. We define a patch $\mathcal{P}[\boldsymbol{z}^{(\ell-1)}](p_i, p_j) \in \{1, \ldots, I^{(\ell)}\} \times \{1, \ldots, J^{(\ell)}\}$ as the slice of the input with indices $c = 1, \ldots, K^{(\ell)}, i = $ (all channels) and $(i, j) \in \{p_i, \ldots, p_i + I_C^{(\ell)}\} \times \{p_i, \ldots, p_i + J_C^{(\ell)}\}$, hence a patch starting at position $(p_i, p_j)$ and of same shape as the filters.

Apodizing a signal in general corresponds to applying an apodization function (or windowing function)Weisstein (2002) $h$ onto it via an Hadamard product. Let define the $2D$ apodized functions $h$ : $\Omega(I_C^{(\ell)}, J_C^{(\ell)}) \to \mathbb{R}^+$ with $\Omega(I_C^{(\ell)}, J_C^{(\ell)}) = \{1, \ldots, I_C^{(\ell)}\} \times \{1, \ldots, J_C^{(\ell)}\}$ and where we remind that $(I_C^{(\ell)}, J_C^{(\ell)})$ is the spatial shape of the convolutional filters. Given a function $h$ such that $\sum_{u \in \Omega(I_C^{(\ell)}, J_C^{(\ell)})} h(u) = 1$ one can represent an input by summing the apodized patches as in

$$[\boldsymbol{z}^{(\ell)}]_{k,i,j} = \sum_{(p_i, p_j) \in \{i - I_C^{(\ell)}, \ldots, i\} \times \{j - J_C^{(\ell)}, \ldots, j\}} \mathcal{P}[\boldsymbol{z}^{(\ell-1)}](p_i, p_j) \odot h. \tag{17}$$

The above highlights the ability to treat an input via its collection of patches with the condition to apply the defined apodization function. With the above, we can demonstrate how minimizing the per patch reconstruction loss leads to minimizing the overall input modeling

$$0 \leq \| \sum_{i,j} (h \odot \mathcal{P}[\boldsymbol{z}^{(\ell)}](i, j) - [W^{(\ell)}]_{t^{(\ell)}(i,j)}) \|^2 \leq \sum_{i,j} \| h \odot \mathcal{P}[\boldsymbol{z}^{(\ell)}](i, j) - [W^{(\ell)}]_{t^{(\ell)}(i,j)} \|^2, \tag{18}$$

which represents the internal modeling of the factorial model applied across filters and patches. As a result, when performing the per position minimization one minimizes an upper bound which ultimately reaches the global minimum as

$$\| \mathcal{P}[\boldsymbol{z}^{(\ell-1)}](p_i, p_j) - \mathcal{P}[\hat{\boldsymbol{z}}^{(\ell-1)}](p_i, p_j) \|^2 \to 0 \implies \| \boldsymbol{z}^{(\ell-1)} - \sum_{(p_i, p_j)} \mathcal{P}[\boldsymbol{z}^{(\ell-1)}](p_i, p_j) \|^2 = 0. \tag{19}$$

## C    INTERPRETATION: INITIALIZATION AND INPUT SPACE PARTITIONING

The GMM formulation and related inference also allows interpretation of the internal layer parameters. First we demonstrate how the region prior $\pi^{(\ell)}$ is affected by the layer parameters especially at initialization. Then we highlight how our result allows to generalize the input space partitioning results from Balestriero & Baraniuk (2018b;a).

**Region Prior.**    The region prior of the GMM-MASO model $[\pi^{(\ell)}]_{k,.}$ (recall Thm. 2) depends on the bias and norm of the layer weight as $[\pi^{(\ell)}]_{k,.} \propto e^{[B^{(\ell)}]_{k,r} + \frac{1}{2}\|[A^{(\ell)}]_{k,r,.}\|^2}$. We can study how this region prior looks like at initialization. At initialization, common practice uses $[B^{(\ell)}]_{k,r} = 0, \forall k, r$ and $[A^{(\ell)}]_{k,r,d} \sim \mathcal{N}(0, (v^{(\ell)})^2)$. This bias initialization leads to a cluster prior probability proportional to the norm of the weights. For example, the case of absolute value leads to $E(\|[A^{(\ell)}]_{k,1,.}\|^2) = E(\|[A^{(\ell)}]_{k,2,.}\|^2)$ and thus uniform prior as $E([\pi^{(\ell)}]_{k,.}) = (0.5, 0.5)^T$ for any initialization standard deviation $v^{(\ell)}$. On the other hand, ReLU has always $\|[A^{(\ell)}]_{k,2,.}\|^2 = 0$ and $E\left(\|[A^{(\ell)}]_{k,r,.}\|^2\right) = D^{(\ell)}(v^{(\ell)})^2$. If one uses Xavier initialization Glorot & Bengio (2010) then $D^{(\ell)}(v^{(\ell)})^2 = 1$ and we thus have as prior probability $[\pi^{(\ell)}]_{k,.} \approx (0.62, 0.38)^T$. THe latter slightly favors the inactive state of the ReLU and thus sparser activations. In general, the smaller $v^{(\ell)}$ is, the more the region prior will favor inactive state of the ReLU.

**Input Space Partitioning.**    We now generalize the ability to study the input space partitioning which was before limited to the special case of $[B^{(\ell)}]_{k,r} = -\frac{1}{2}\|[A^{(\ell)}]_{k,r,.}\|^2$ (recall Prop. 1). Studying the input space partition is crucial as the MASO property implies that for each input region, an observation is transformed via a simple linear transformation. However, deriving insights on that is the actual partition is cumbersome as analytical formula are impractical and one thus has to probe the input space and record the observed VQ for each point to estimate the input space partitioning. We are now able to derive some clear links between the MASO partition and standard models which will allows much more efficient computation of the input space partitions.

**Corollary 2.** *A MASO with arbitrary parameters $[A^{(\ell)}]_{k,r,.}, [B^{(\ell)}]_{k,r}$ has an input space partitioning being the same as a GMM with parameters from Thm. 2.*

This augments previous study of the MASO input space partitioning only related to k-mean (recall Prop. 1) which required specific bias values.

## D  DEEP NETWORK TOPOLOGIES AND DATASETS

We first present the topologies used in the experiments except for the notation ResNetD-W which is the standard wide ResNet based topology with depth $D$ and width $W$. We thus have the following network architectures for smallCNN and largeCNN:

largeCNN

```
Conv2DLayer(layers[-1],96,3,pad='same')
Conv2DLayer(layers[-1],96,3,pad='same')
Conv2DLayer(layers[-1],96,3,pad='same',stride=2)
Conv2DLayer(layers[-1],192,3,pad='same')
Conv2DLayer(layers[-1],192,3,pad='same')
Conv2DLayer(layers[-1],192,3,pad='same',stride=2)
Conv2DLayer(layers[-1],192,3,pad='valid')
Conv2DLayer(layers[-1],192,1)
Conv2DLayer(layers[-1],10,1)
GlobalPoolLayer(layers[-1],2)
```

where the Conv2DLayer(layers[-1],192,3,pad='valid') denotes a standard 2D convolution with 192 filters of spatial size $(3,3)$ and with valid padding (no padding).

## E  PROOFS

### E.1  THEOREM 2

*Proof.* The log-probability of the model corresponds to

$$
\begin{aligned}
[t^{(\ell)}]_k ={}& \arg\max_r \langle [A^{(\ell)}]_{k,r,.}, z^{(\ell-1)} \rangle + [B^{(\ell)}]_{k,r} \\
={}& \arg\max_r \langle [A^{(\ell)}]_{k,r,.}, z^{(\ell-1)} \rangle + [B^{(\ell)}]_{k,r} + \frac{1}{2}\|[A^{(\ell)}]_{k,r,.}\|^2 - \frac{1}{2}\|[A^{(\ell)}]_{k,r,.}\|^2 \\
={}& \arg\max_r \langle [A^{(\ell)}]_{k,r,.}, z^{(\ell-1)} \rangle + [B^{(\ell)}]_{k,r} + \frac{1}{2}\|[A^{(\ell)}]_{k,r,.}\|^2 \\
& - \log\left( \sum_r e^{[B^{(\ell)}]_{k,r}+\frac{1}{2}\|[A^{(\ell)}]_{k,r,.}\|^2} \right) - \frac{1}{2}\|[A^{(\ell)}]_{k,r,.}\|^2 \\
={}& \arg\max_r \langle [A^{(\ell)}]_{k,r,.}, z^{(\ell-1)} \rangle + \log\left( e^{[B^{(\ell)}]_{k,r}+\frac{1}{2}\|[A^{(\ell)}]_{k,r,.}\|^2} \right) \\
& - \log\left( \sum_r e^{[B^{(\ell)}]_{k,r}+\frac{1}{2}\|[A^{(\ell)}]_{k,r,.}\|^2} \right) - \frac{1}{2}\|[A^{(\ell)}]_{k,r,.}\|^2 \\
={}& \arg\max_r \langle [A^{(\ell)}]_{k,r,.}, z^{(\ell-1)} \rangle + \log\left( \frac{e^{[B^{(\ell)}]_{k,r}+\frac{1}{2}\|[A^{(\ell)}]_{k,r,.}\|^2}}{\sum_r e^{[B^{(\ell)}]_{k,r}+\frac{1}{2}\|[A^{(\ell)}]_{k,r,.}\|^2}} \right) - \frac{1}{2}\|[A^{(\ell)}]_{k,r,.}\|^2 \\
={}& \arg\max_r \ \log\left( p(x|r)p(r) \right) - \frac{1}{2}\|z^{(\ell-1)}\|^2 \\
={}& \arg\max_r \ p(x|r)p(r)
\end{aligned}
$$

We also remind the reader that $\arg\max_r p(z^{(\ell-1)}|r)p(r) = \arg\max_r \log(p(z^{(\ell-1)}|r)p(r))$. Based on the above it is straightforward to derive (5) from the above. □

### E.2 Entropy Regularized Optimization

*Proof.* We are interested into the following optimization problem:

$$[t^{(\ell)*}]_k = \underset{q[\ell,k]}{\arg\max} \, F(q[\ell,k], \Theta) = \underset{q[\ell,k]}{\arg\max} \, E_q[\log(p(z^{(\ell-1)}|[t^{(\ell)}]_k)p([t^{(\ell)}]_k))] + H([t^{(\ell)}]_k)$$

$$= \underset{u^{(\ell)} \in \triangle_R}{\arg\max} \left( \sum_r [u^{(\ell)}]_{k,r} [\frac{-1}{2\sigma^2} \|z^{(\ell-1)} - \mu_r\|^2 + \log(\pi_r)] - \sum_r [u^{(\ell)}]_{k,r} \log([u^{(\ell)}]_{k,r}) \right).$$

We now use the KKT and Lagrange multiplier to optimize the new loss function (per $k$) including the equality constraint

$$\mathcal{L}(u) = \sum_r [u]_r [\frac{-1}{2\sigma^2} \|z^{(\ell-1)} - \mu_r\|^2 + \log(\pi_r)] - \sum_r [u]_r \log([u^{(\ell)}]_r) + \lambda(\sum_r [u]_r - 1)$$

Due to the strong duality we can directly optimize the primal and dual problems and solve jointly all the partial derivatives to 0. We thus obtain by denoting $A_r := [\frac{-1}{2\sigma^2} \|z^{(\ell-1)} - \mu_r\|^2 + \log(\pi_r)]$

$$\frac{\partial \mathcal{L}}{\partial [u]_p} = A_p - \log([u]_p) - 1 + \lambda, \forall p$$

$$\frac{\partial \mathcal{L}}{\partial \lambda} = \sum_r [u]_r - 1$$

we can now set the derivatives to 0 and see that this leads to $[u]_p = e^{A_p - 1 + \lambda}, \forall p$. We can now sum over $p$ to obtain

$$[u]_p = e^{A_p - 1 + \lambda}, \forall p \implies \sum_p [u]_p = \sum_p e^{A_p - 1 + \lambda}$$

$$\implies 1 = \sum_p e^{A_p - 1 + \lambda}$$

$$\implies 1 = e^\lambda \sum_p e^{A_p - 1}$$

$$\implies 0 = \lambda + \log(\sum_p e^{A_p - 1})$$

which leads to $\lambda = -\log(\sum_p e^{A_p - 1})$. Plugging this back into the above equation we obtain

$$[u]_p = e^{A_p - 1 + \lambda} = \frac{e^{A_p - 1}}{\sum_p e^{A_p - 1}} = \frac{e^{A_p}}{\sum_p e^{A_p}}$$

$\square$

### E.3 Theorem 3

For the proof of Theorem 3 please refer to the proof in E.2 by applying the convex combination with coefficients $\beta$.

### E.4 Theorem 4

*Proof.* The proof to demonstrate this inference and VQ equality is essentially the same as the one of GMM-MASO (E.1) with addition of the following first step:

$$\|z^{(\ell-1)} - \sum_{k=1}^{D^{(\ell)}} [W^{(\ell)}]_{k,r,.}\|^2 = \|z^{(\ell-1)}\|^2 - 2 \sum_{k=1}^{D^{(\ell)}} \sum_{r=1}^{R^{(\ell)}} [W^{(\ell)}]_{k,[r_k],.} + \sum_{k=1}^{D^{(\ell)}} \|[W^{(\ell)}]_{k,[r_k],.}\|^2$$

for any configuration $r \in \{1, \dots, R^{(\ell)}\}^{D^{(\ell)}}$. Using the same results we can re-write the independent joint optimization as multiple independent optimization problems. $\square$

### E.5 PROPOSITIONS 3 AND 4

For Proposition 4 using the developed formula one can extend the following proof for max-pooling.

*Proof.*

$$
\begin{aligned}
[\boldsymbol{z}^{(\ell)}(\boldsymbol{x})]_k &= \frac{\langle e^{A^{(\ell)}[k,2],\boldsymbol{z}^{(\ell-1)(\boldsymbol{x})}\rangle+B^{(\ell)}[k,2]}}{1+e^{\langle A^{(\ell)}[k,2],\boldsymbol{z}^{(\ell-1)}(\boldsymbol{x})\rangle+B^{(\ell)}[k,2]}} \times (\langle A^{(\ell)}[k,2],\boldsymbol{z}^{(\ell-1)}(\boldsymbol{x})\rangle+B^{(\ell)}[k,2]) \\
&= \sigma_{\text{sigmoid}}(\langle A^{(\ell)}[k,2],\boldsymbol{z}^{(\ell-1)}(\boldsymbol{x})\rangle+B^{(\ell)}[k,2])(\langle A^{(\ell)}[k,2],\boldsymbol{z}^{(\ell-1)}(\boldsymbol{x})\rangle+B^{(\ell)}[k,2]) \\
&= \sigma_{\text{sigmoid}}(\langle [\boldsymbol{C}^{(\ell)}]_{k,.},\boldsymbol{z}^{(\ell-1)}(\boldsymbol{x})\rangle+[b_C^{(\ell)}]_k) \times (\langle [\boldsymbol{C}^{(\ell)}]_{k,.},\boldsymbol{z}^{(\ell-1)}(\boldsymbol{x})\rangle+[b_C^{(\ell)}]_k) \quad (20)
\end{aligned}
$$

with 1 for the first region exponential as $e^{\langle A^{(\ell)}[k,2],\boldsymbol{z}^{(\ell-1)}(\boldsymbol{x})\rangle+B^{(\ell)}[k,1]} = e^0 = 1$ and the last line demonstrating the case where ReLU activation and convolution was the internal layer configuration for illustrative purposes. □

### E.6 PROPOSITIONS 5 AND 6

*Proof.*

$$
\begin{aligned}
p([t^{(\ell)}]_k=1|\boldsymbol{z}^{(\ell-1)}) &= \frac{p(\boldsymbol{z}^{(\ell-1)}|[t^{(\ell)}]_k=1)p([t^{(\ell)}]_k=1)}{p(\boldsymbol{z}^{(\ell-1)})} \\
&= \frac{p(\boldsymbol{z}^{(\ell-1)}|[t^{(\ell)}]_k=1)p([t^{(\ell)}]_k=1)}{p(\boldsymbol{z}^{(\ell-1)}|[t^{(\ell)}]_k=0)p([t^{(\ell)}]_k=0)+p(\boldsymbol{z}^{(\ell-1)}|[t^{(\ell)}]_k=1)p([t^{(\ell)}]_k=1)} \\
&= \frac{e(-\frac{\|\boldsymbol{z}^{(\ell-1)}-[\boldsymbol{C}^{(\ell)}]_{k,.}\|^2}{2})\frac{e(\frac{1}{2}\|[\boldsymbol{C}^{(\ell)}]_{k,.}\|^2+[B^{(\ell)}]_{k,1})}{1+e(\frac{1}{2}\|[\boldsymbol{C}^{(\ell)}]_{k,.}\|^2+[B^{(\ell)}]_{k,1})}}{e(-\frac{\|\boldsymbol{z}^{(\ell-1)}\|^2}{2})\frac{1}{1+e(\frac{1}{2}\|[\boldsymbol{C}^{(\ell)}]_{k,.}\|^2+[B^{(\ell)}]_{k,1})}+e(-\frac{\|\boldsymbol{z}^{(\ell-1)}-[\boldsymbol{C}^{(\ell)}]_{k,.}\|^2}{2})\frac{e(\frac{1}{2}\|[\boldsymbol{C}^{(\ell)}]_{k,.}\|^2+[B^{(\ell)}]_{k,1})}{1+e(\frac{1}{2}\|[\boldsymbol{C}^{(\ell)}]_{k,.}\|^2+[B^{(\ell)}]_{k,1})}} \\
&= \frac{e(-\frac{\|\boldsymbol{z}^{(\ell-1)}-[\boldsymbol{C}^{(\ell)}]_{k,.}\|^2}{2})e(\frac{1}{2}\|[\boldsymbol{C}^{(\ell)}]_{k,.}\|^2+[B^{(\ell)}]_{k,1})}{e(-\frac{\|\boldsymbol{z}^{(\ell-1)}\|^2}{2})+e(-\frac{\|\boldsymbol{z}^{(\ell-1)}-[\boldsymbol{C}^{(\ell)}]_{k,.}\|^2}{2})e(\frac{1}{2}\|[\boldsymbol{C}^{(\ell)}]_{k,.}\|^2+[B^{(\ell)}]_{k,1})} \\
&= \frac{e(\langle \boldsymbol{z}^{(\ell-1)},[\boldsymbol{C}^{(\ell)}]_{k,.}\rangle+[B^{(\ell)}]_{k,1})}{1+e(\langle \boldsymbol{z}^{(\ell-1)},[\boldsymbol{C}^{(\ell)}]_{k,.}\rangle+[B^{(\ell)}]_{k,1})} \\
&= \sigma_{\text{sigmoid}}([u^{(\ell)}]_k).
\end{aligned}
$$

While this is direct for sigmoid DNs, the use of hyperbolic tangent requires to reparametrize the current and following layer weights and biases to represent the shifting scaling as in $\boldsymbol{C}^{(\ell)} := 2\boldsymbol{C}^{(\ell)}$ and $\boldsymbol{C}^{(\ell+1)} := 2\boldsymbol{C}^{(\ell+1)}, b_C^{(\ell+1)} := b_C^{(\ell+1)} - 1$ with $\boldsymbol{C}$ replaced by $W$ for fully connected operators. □

