# OpenReview forum: "From Hard to Soft: Understanding Deep Network Nonlinearities via Vector Quantization and Statistical Inference"
_ICLR.cc/2019/Conference_

### Official Review · AnonReviewer2 · 2018-11-02
**This paper extends the max-affine spline operator (MISO) interpretation of a class of deep neural networks to cover a wider class of activation functions, namely the sigmoid, hyperbolic tangent and softmax. The authors also use the formulation to create a family of models that interpolates between hard and soft non-linearities.**

**Rating:** 7
**Confidence:** 5

**Review:**

Interesting work, extending previous work by Balestriero and Baraniuk in a relevant and non-trivial direction. The presentation could be cleaner and clearer,

The paper contains solid work and contributes to an interesting perspective/interpretation of deep networks. The presentation is reasonably clear, although somewhat cluttered by a large number of subscripts and superscripts, which could be avoided by using a more modular formulation; e.g., in equation (1), when referring to a specific layer l, the superscript l can be dropped as it adds no useful information. By the way, when l is first used, just before equation (1), it is undefined, although the reader can guess what it stands for.

It is not clear why $[\pi^{(l)}]_{k,t}$ is defined after equation (5), as these quantities are not mentioned in Theorem 2. Another confusion issue is that it is not clear if the assumption made in Proposition 1 concerning is only valid there of if it is assued to hold elsewhere in the paper.

Proposition 2 is simply a statement of the well-known relationship between between soft-max (a.k.a. logistic regression) and the maximum entropy principle (see, for example, http://www.win-vector.com/dfiles/LogisticRegressionMaxEnt.pdf).

---

> ### Author Response · Authors · 2018-11-09
> **Answer to Reviewer2**
>
> We thank the reviewer for their constructive comments. We have revised the manuscript accordingly.
>
> We have simplified the background (Section 2) by removing the superfluous l (layer) superscript.  This reworking clarifies the definition and operation of the MASO.
>
> We have defined  $[\pi^{(l)}]_{k,t}$ in the early part of Theorem 2 and then derived (5).
>
> The assumption on the bias value needed for Proposition 1 is indeed only needed for that particular result.  We have highlighted this (i) in the second paragraph following Proposition 1 and (ii) in the sentence immediately after Theorem 2.
>
> We have highlighted that Proposition 2 is a standard result (and added references) and motivated its presence in order to unify all the different VQs under a single optimization problem. Adding an Entropy regularization to the original optimization problem then enables us to interpolate between hard,soft and linear VQ.

---

### Official Review · AnonReviewer3 · 2018-11-03
**Somewhat incremental work, but well posited and written.**

**Rating:** 6
**Confidence:** 4

**Review:**

This work extends the applicability of the spline theory of deep networks explored in previous works of Balestriero/ Baraniuk. The previous works setup DNs as layer-wise max-affine spline operators (MASOs) and recovers several non-linearities practically used as special cases of these MASOs. The previous works already recover RELU variants and some downsampling operators that the current submission characterizes as "hard" quantization.

The major contribution of this work is extending the application to "soft" quantization that recovers several new non-linear activations such as soft-max. It is well-known that the k-means algorithm can be considered as a run of an EM algorithm to recover the mean parameters of a gaussian mixture model. The "hard" to "soft" transformation, and any interpolation in between follows from combining this insight with the previous works. As such there isnt a major technical contribution imho in this work. Furthermore, the presented orthogonalization for easier inference has been used before in many works, some of which this submission also cites, most importantly in the previous work of Balestriero/ Baraniuk that this submission extends.

Nevertheless there is value in novel results that may follow from previous works in a straightforward but non-trivial fashion, as long as it is well-presented and thoroughly researched and implication well-highlighted. This paper does that adequately, so I will suggest weak accept. Furthermore, this work could spark interesting future works and fruitful discussions at the ICLR. It is well-written and the experimental evaluation is adequate.

I would suggest a couple of ways to possibly improve the exposition. The paper is somewhat notation heavy. When considering single layers, the superscript for the layer could be dropped in favor of clarity. I would suggest moving the definition of MASOs to the main text, and present Proposition 8 in some form in the main text as well. To a reader not familiar with previous works, or with splines, this could be helpful. Use of orthogonalization could be highlighted not just a tool for tractability but also regularization. For inference on GMMs, it corresponds to a type of variational inference, which could be mentioned.

---

> ### Author Response · Authors · 2018-11-09
> **Answer to Reviewer3**
>
> We thank the reviewer for their constructive comments. We respond to each below in detail.
>
> TECHNICAL CONTRIBUTIONS
> We briefly review our four primary technical contributions.
> [C1] We extend the deterministic max-affine spline operator (MASO) framework for deep networks (DNs) developed in (Balestriero & Baraniuk, ICML2018) to a probabilistic Gaussian mixture model (GMM).
> [C2] We extend the deterministic vector quantization (VQ) spline partition of the MASO framework to a probabilistic, soft VQ that enables us to derive from first principles and unify most of the known DN nonlinearities, including nonlinear and nonconvex ones such as the softmax and sigmoid gated linear unit.
> [C3] By interpolating between hard and soft inference, we derive a new class of beta-VQ activation functions. In particular, a beta-VQ version of the hard ReLU activation is the “Swish” nonlinearity, which offers state-of-the-art performance in a range of computer vision tasks but was proposed ad hoc through experimentation.
> [C4] We rigorously prove that orthogonal filters endow a DN with an attractive inference capability. Orthogonal filters enable a DN to perform efficient, tractable, jointly optimal VQ inference across all units in a layer. This is in contrast to non-orthogonal DNs, which support optimal VQ only on a per-unit basis. Previous works have studied orthogonality only empirically.
>
> ORTHOGONALIZATION
> As noted in contribution [4] above, orthogonalization has already been applied in deep learning, but it has typically been applied ad hoc with little to no theoretical justification. In our paper, we have justified orthogonalization from the novel point of view of inferring the VQ partition of each of the unit outputs in a DN layer. In a standard DN, each unit output computation is performed independently from the other units. This absence of ‘’lateral connections’’ can lead to two problematic situations: on the one hand redundant information in a feature map or on the other hand incomplete representation of the input. We demonstrate that an elegant solution to both problems is to enforce orthogonality. We have added a statement after Theorem 4 regarding how orthogonalization has potential applications of independent interest outside of deep learning, for example in factorial GMMs and HMMs.
>
> FUTURE DIRECTIONS AND DISCUSSIONS
> We agree with the reviewer that our hard/soft VQ perspective opens up many new directions to both understand and improve DNs. Here are several new directions that we could discuss further in the revised paper or at the conference:
> [F1] VQ penalization: Given our explicit (and differentiable) formulas for the soft VQ, we can derive new kinds of penalties to apply during learning.  For example, we could penalize an overconfident-VQ (as measured by the joint likelihood of the unit VQ representation of the layer input), which is symptomatic of over-fitting.
> [F2] Leaning new activation functions: The state-of-the-art Swish nonlinearity has learnable parameter that enables it to range from ReLU to sigmoid gated linear unit to linear. We can further augment this parametrization to enable us to reach the sigmoid unit as well. This will enable us to use learning experiments to investigate the conjecture that ReLU like nonlinearities are best for early DN layers while sigmoid-like nonlinearities are best for later layers.
> [F3] We can use the VQ and the per-unit VQ-based likelihood to create DNs that detect outliers and perform model selection.
> [F4] Alternative soft-VQ regularization: Replacing the Shannon Entropy regularization in (7) with a different penalty could yield new classes of nonlinear activation functions.

---

### Official Review · AnonReviewer1 · 2018-11-09
**Logical continuation of existing work**

**Rating:** 6
**Confidence:** 3

**Review:**

At the core of this paper is the insight from [1] that a neural network layer constructed from a combination of linear, piecewise affine and convex operators can be interpreted as a max-affine spline operator (MASO). MASOs are directly connected to vector quantization (VQ) and K-means clustering, which means that a deep network implicitly constructs a hierarchical clustering of the training data during learning. This paper now substitutes VQ with probabilistic clustering models (GMMs) and extends the MASO interpretation of a wider range of possible operations in deep neural networks (sigmoidal activation functions, etc.).

Given the detailed treatment of MASOs in [1], this paper is a logical continuation of this approach. As such, it may seem only incremental, but I would consider it as an important piece to ensure a solid foundation of the 'MASO-view' on deep neural networks.

My main criticism is with respect to the quality and clarity of the presentation. Without reading in detail [1] it is very difficult to understand the presented work here. Moreover, compared to [1], a lot of explanatory content is missing, e.g. [1] had nice visualisations of the resulting partitioning on toy data.

Clearly, this work and [1] belong together in a larger form (e.g. a journal article), I hope that this is considered by the authors.

---

> ### Comment · AnonReviewer1 · 2018-11-09
> **Adding cite for [1]**
>
> [1] Mad Max: Affine Spline Insights into Deep Learning https://arxiv.org/abs/1805.06576

---

> ### Author Response · Authors · 2018-11-09
> **Answer to Reviewer1**
>
> We thank the reviewer for their constructive comments. We agree that our soft-VQ extension is an important piece of the puzzle that is necessary to ensure a solid foundation of the 'MASO-view' of deep neural networks.
>
> Regarding the clarity of presentation, we agree that our streamlined treatment of the MASO background, while self-contained, is quite terse. The reason is very short page limit allowed for the submission. We hope that the reader will find our new results compelling enough that they will refer to [1] for additional background information and insights.
>
> Regarding the experiments and visualization, we also had to make hard choices due to space limitations. We decided that repeating visualizations from [1] using a Soft-VQ partitioning would be less useful than a detailed derivation and explanation of the internal Hard/Soft/Beta-VQ processes and how they lead to new nonlinearities. We certainly plan to include many more visualizations in our conference presentation, should the paper be accepted.
>
> Finally, we feel that our extension of the deterministic MASO framework is more than incremental, since it opens the door to a range of new applications, improvements, and theoretical questions that go far beyond the scope of [1]. For some examples, please see our reply to Reviewer 3.
>
> [1] Mad Max: Affine Spline Insights into Deep Learning https://arxiv.org/abs/1805.06576

---

### Meta-Review · Area_Chair1 · 2018-12-13
**Nice piece of work**

**Confidence:** 4
**Recommendation:** Accept (Poster)

**Metareview:**

Dear authors,

All reviewers liked your work. However, they also noted that the paper was hard to read, whether because of the notation or the lack of visualization.

I strongly encourage you to spend the extra effort making your work more accessible for the final version.